# Association Between Pentraxins and Obesity in Prediabetes and Newly Diagnosed Type 2 Diabetes Mellitus Patients

**DOI:** 10.3390/ijms26083661

**Published:** 2025-04-12

**Authors:** Roxana-Viorela Ahrițculesei, Lidia Boldeanu, Daniel Cosmin Caragea, Ionela Mihaela Vladu, Diana Clenciu, Adina Mitrea, Anca Marilena Ungureanu, Constantin-Cristian Văduva, Anda Lorena Dijmărescu, Alin Iulian Silviu Popescu, Mohamed-Zakaria Assani, Mihail Virgil Boldeanu, Cristin Constantin Vere

**Affiliations:** 1Doctoral School, University of Medicine and Pharmacy of Craiova, 200349 Craiova, Romania; roxana.blendea@gmail.com; 2Department of Microbiology, Faculty of Medicine, University of Medicine and Pharmacy of Craiova, 200349 Craiova, Romania; anca.ungureanu@umfcv.ro; 3Department of Nephrology, Faculty of Medicine, University of Medicine and Pharmacy of Craiova, 200349 Craiova, Romania; caragea.daniel87@yahoo.com; 4Department of Diabetes, Nutrition and Metabolic Diseases, Faculty of Medicine, University of Medicine and Pharmacy of Craiova, 200349 Craiova, Romania; ionela.vladu@umfcv.ro (I.M.V.); dianaclenciu@yahoo.com (D.C.); ada_mitrea@yahoo.com (A.M.); 5Department of Obstetrics and Gynecology, Faculty of Medicine, University of Medicine and Pharmacy of Craiova, 200349 Craiova, Romania; cristian.vaduva@umfcv.ro (C.-C.V.); lorenadijmarescu@yahoo.com (A.L.D.); 6Department of Internal Medicine, University of Medicine and Pharmacy of Craiova, 200349 Craiova, Romania; alin.popescu@umfcv.ro; 7Department of Immunology, Faculty of Medicine, University of Medicine and Pharmacy of Craiova, 200349 Craiova, Romania; mihail.boldeanu@umfcv.ro; 8Department of Gastroenterology, University of Medicine and Pharmacy of Craiova, 200349 Craiova, Romania; vere_cristin@yahoo.com

**Keywords:** type 2 diabetes mellitus, pentraxin 3, high-sensitivity C-reactive protein, tumor necro-sis factor alpha, interleukin 6, MCVL, mean corpuscular volume-to-lymphocyte ratio, IIC, cumulative inflammatory index, obesity

## Abstract

Systemic inflammation has an important role in the prognosis and progression of many chronic diseases, including diabetes (T2DM). This retrospective study aimed to evaluate inflammatory status by determining the serum inflammatory biomarkers (PTX3, hs-CRP, TNF-α, and IL-6) and new indices, like the mean corpuscular volume (MCV) to lymphocyte ratio (MCVL) and cumulative inflammatory index (IIC), in a cohort of patients with prediabetes (PreDM) and newly diagnosed T2DM. We also wanted to assess the association with clinical parameters and different obesity-related indices, to identify possible correlations and to evaluate the diagnostic accuracy of the biomarkers using ROC curve analysis. In this study, we included 60 patients diagnosed with T2DM and 30 patients with PreDM. The ELISA method was applied. Elevated PTX3, hs-CRP, TNF-α, and IL-6 levels were found in T2DM patients compared to preDM patients. An independent relationship was found between PTX3, hs-CRP, and different obesity-related indices in patients with preDM and T2DM. The MCVL index exhibited an inverse trend proportional to the rising levels of HbA1c in the T2DM group. Spearman’s analysis revealed in the T2DM group that the PTX3 values correlated much better with IIC (rho = 0.445, *p*-value = 0.014) and MCVL (rho = 0.338, *p*-value = 0.048). Hs-CRP values expressed moderate-to-weak correlations with IIC and MCVL in both groups. Additionally, ROC analysis showed that the PTX3 (AUC was 0.720; *p* = 0.003; cut-off value 1888.00 pg/mL, with 67.60% sensitivity and 73.30% specificity) and MCVL index (AUC was 0.677; *p* = 0.047; cut-off value 39.60, with 63.30% sensitivity and 66.70% specificity) have a good, accurate diagnosis compared with IL-6 (AUC was 0.866; *p* < 0.0001; cut-off value 40.30 pg/mL, with 100.00% sensitivity and 60.00% specificity). IIC showed 61.70% sensitivity and 60.00% specificity, with an AUC of 0.572, *p* = 0.027 and a cut-off value of 2.35. PTX3 and MCVL can serve as independent predictor factors in the inflammatory status in preDM and T2DM patients, supporting their potential as biomarkers for T2DM management and future research.

## 1. Introduction

Diabetes mellitus (DM) represents a considerable global health challenge, projected to impact 537 million individuals worldwide in the year 2021. By 2030, this figure is anticipated to rise to 643 million individuals, accounting for 11.3% of the adult population, and by 2045, the number is expected to reach 783 million individuals, or 12.2% [1,2]. The prevalence of diabetes is increasing, which significantly affects morbidity and mortality rates [3]. Individuals diagnosed with diabetes have a two to three-fold increased risk of mortality from all causes, as well as from cardiovascular diseases, in comparison to those without diabetes [4].

Diabetes represents one of numerous chronic diseases whose prognosis and trajectory are markedly affected by systemic inflammation. Inflammation plays a critical role in determining the progression of the disease, as well as the clinical outcomes for diabetic patients. Numerous studies conducted over recent decades have established a positive correlation between diabetes and suboptimal levels of inflammatory markers and mediators [C-reactive protein (CRP), interleukin-1beta (IL-1b), IL-6, tumor necrosis factor-alpha (TNF-α), leptin, adiponectin, and adipocytokine] [5,6,7,8,9,10,11,12,13]. Type 2 diabetes mellitus (T2DM), insulin resistance, hypertension, metabolic syndrome, nonalcoholic fatty liver disease, and cardiovascular disease are all characterized by persistent low-grade systemic inflammation. Therefore, obesity is considered the principal factor linking inflammation to chronic comorbidities [14,15,16,17].

Excessive adipose tissue accumulation in the body is a defining characteristic of obesity. Adipose tissue is now recognized as an endocrine organ that performs immune functions [18]. In patients with obesity, the quantity and functionality of immune cells within the adipose tissue are notably modified due to excessive fat accumulation. While there is a reduction in the number of eosinophils and certain subgroups of T lymphocytes, there is an increase in macrophages, mast cells, neutrophils, T lymphocytes, and B lymphocytes. These alterations are associated with both systemic and local inflammation [19].

In recent years, hematological indices, functioning as composite biomarkers, have been recognized for their enhanced capacity to more accurately reflect both innate and adaptive immune responses. This recognition is attributed to their superior sensitivity, accuracy, and availability in comparison to traditional serological indicators (erythrocyte sedimentation rate (ESR), fibrinogen (FIB), CRP, and white blood cell count (WBC)), such as the neutrophil-to-lymphocyte ratio (NLR), which emphasizes lymphopenia and leukocytosis during the early stages of inflammation [20,21,22,23,24,25,26,27]; monocyte-to-lymphocyte ratio (MLR), which influences the body’s immune health; a drop in its value, suggesting that the host’s immune system is malfunctioning [26]; platelet-to-lymphocyte ratio (PLR), which is a marker of inflammation for immune-mediated, metabolic, and prothrombotic disorders [26,27]; the aggregate index of systemic inflammation (AISI) ((neutrophils × monocytes × platelets)/lymphocytes) [26,28,29,30], also known as the systemic immune-inflammation response index (SIIRI) or pan-immune-inflammation value (PIV); the systemic immune-inflammation index (SII) ((neutrophils × platelets)/lymphocytes) [27,28,29,30,31,32,33]; and additionally, novel hematological indices, like the mean corpuscular volume (MCV) to lymphocyte (MCVL) ratio [34,35,36], and the cumulative inflammatory index (IIC) ((mean corpuscular volume × width of erythrocyte distribution × neutrophils)/(lymphocytes × 1000)) [34,35,36,37].

The C-reactive protein (CRP) is classified as a member of the short pentraxin (PTX) family. This acute-phase protein is synthesized by hepatocytes in response to microbial invasion or tissue injury, with levels typically increasing during the early stages of tissue damage. While both CRP and PTX3 fall under the category of acute-phase proteins within the pentraxin family, they exhibit distinct biological characteristics. PTX3, in particular, is characterized by a longer chain structure. The high-sensitivity C-reactive protein (hs-CRP) and PTX3 can be utilized synergistically to examine the relevance of various disorders. Notably, a diverse array of tissue cells (endothelial cells, fibroblasts, monocytes, and adipocytes) can release PTX3 when stimulated by pro-inflammatory factors [38,39].

PTX3 and hs-CRP have been linked in recent years to the development and progression of diabetic nephropathy (DN) [40,41], the pathogenesis of diabetic kidney disease (DKD) [42], the development and progression of diabetic retinopathy (DR) [43], prediabetes (PreDM) and T2DM in obese patients with nonalcoholic fatty liver disease [44], and the risk of cardiovascular diseases in both the general population and T2DM patients [45].

Research aimed at investigating the relationship between systemic immune inflammation status, as indicated by inflammatory biomarkers (PTX3, hs-CRP, TNF-α, and IL-6) or established hematological markers (NLR, MLR, PLR, AISI, SII) alongside novel markers, specifically MCVL and IIC, and obesity (various obesity-related indices) in patients with PreDM and newly diagnosed T2DM, as documented in both the international and Romanian specialized literature, remains scarce.

This study retrospectively examined patients with PreDM and newly diagnosed T2DM to assess their immune and inflammatory levels by quantifying specific biomarkers in their blood, including PTX3, hs-CRP, TNF-α, and IL-6, as well as hematological indices such as NLR, MLR, PLR, AISI, SII, MCVL, and IIC. This cohort was derived from data obtained from two clinical hospitals in Dolj County, Romania. Furthermore, we sought to evaluate the relationship between these parameters and clinical, as well as biochemical measures, in conjunction with various obesity-related indices, to discern potential correlations among them. The third primary objective of the study was to assess the diagnostic accuracy of PTX3, hs-CRP, TNF-α, IL-6, NLR, MLR, PLR, AISI, SII, MCVL, and IIC as individual early predictive indicators and to determine the inflammatory status in patients with and without T2DM, by employing a receiver operating characteristic (ROC) curve analysis.

This analysis aimed to provide a comprehensive understanding of how these biomarkers interact with obesity and diabetes, potentially leading to improved diagnostic and therapeutic strategies. By identifying specific patterns and correlations, this study hopes to enhance early detection methods for T2DM and offer insights into its underlying inflammatory mechanisms. Ultimately, the findings could pave the way for personalized treatment approaches that target not only the symptoms of these conditions but also their root causes, thereby improving patient outcomes. Continued research in this area is essential for developing effective interventions that can mitigate the growing prevalence of obesity and diabetes worldwide.

## 2. Results

### 2.1. Demographic Characteristics of Patients with Prediabetes and Diabetes

As presented in Table 1, in this study, we included 60 patients diagnosed with T2DM, aged between 43 and 64 years, with a mean ± standard deviation (SD) age of 53.43 ± 5.46, consisting of 28 males and 32 females. In the control group, the PreDM group, we found that the mean ± SD age was 49.87 ± 7.68 and females predominated with a percentage of 56.67. Thus, a statistically significant difference was observed regarding age (*p* = 0.013), but not in the case of gender (χ^2^(1) = 0.089, *p* = 0.765). Regarding where the patients lived, most people in the T2DM and PreDM groups (36 and 18 patients, respectively) were from rural areas. There was no statistically significant difference between the two groups (χ^2^(1) = 0.052, *p* =0.819).

The analysis conducted in this study regarding smoking and drinking histories indicates that a greater proportion of patients with T2DM engage in smoking and alcohol consumption. Nevertheless, the differences observed between the groups under investigation were not statistically significant (*p* = 0.553 and *p* = 0.541, respectively).

Personal history was another significant difference between the T2DM and PreDM groups, as indicated by statistics showing that over 53% of patients had conditions like hypertension, dyslipidemia, or hepatosteatosis. In this context, only diastolic blood pressure (DBP) exhibited statistically significantly higher mean values in the T2DM group.

Concerning the assessed anthropometric parameters, which include height, weight, waist circumference (WC), and hip circumference (HC), as well as various obesity-related indices such as waist-to-height ratio (WHR), waist-to-height ratio (WHtR), body adiposity index (BAI), and body mass index (BMI), statistically significant differences were found between the two groups regarding the mean values for height (*p* = 0.0001), WC (*p* = 0.026), WHR (*p* = 0.006), and BMI (*p* = 0.007). Furthermore, it was noted that 70% of PreDM and 80% of those diagnosed with T2DM were classified as overweight or obese. No statistically significant differences were observed in BMI categories between the two groups.

### 2.2. Laboratory Characteristics of Patients in the Groups Under Study

This study has demonstrated that the mean fasting plasma glucose (FPG), two-hour plasma glucose following a 75 g oral glucose tolerance test (2hPG), and glycosylated hemoglobin A1c (HbA1c) values were significantly elevated (*p* < 0.0001) in the T2DM group as compared to the PreDM group (Table 2).

This study revealed a significant association between elevated serum levels of total cholesterol (TC), total triglycerides (TGs), and low-density lipoprotein cholesterol (LDL-C), and decreased serum levels of high-density lipoprotein cholesterol (HDL-C) in patients with T2DM regarding dyslipidemia.

Regarding kidney function, this study revealed no significant difference in serum blood urea nitrogen (BUN) and creatinine (Crea) levels between the two groups. Additionally, the mean value of the estimated glomerular filtration rate (e-GFR) did not significantly differ between the two groups (T2DM vs. PreDM).

It was observed that serum albumin (ALB) levels exhibited statistically significant differences between the PreDM and T2DM groups.

### 2.3. Comparing the Inflammatory Biomarker Values Between the Studied Groups

Both pentraxins, specifically hs-CRP and PTX3, demonstrated significantly elevated values in individuals with T2DM compared to those in the PreDM group, with respective fold increases of 1.25 and 1.71 (refer to Table 3). Furthermore, the median serum values for tumor necrosis factor alpha (TNF-α) were markedly different between the two cohorts, recorded at 194.00 pg/mL for T2DM versus 84.23 pg/mL for PreDM (*p* = 0.019). Additionally, interleukin-6 (IL-6) levels also exhibited significant variation, measuring 72.34 pg/mL in the T2DM group compared to 29.96 pg/mL in the PreDM group (*p* < 0.0001).

When analyzing complete blood cell counts, our study highlighted that newly diagnosed T2DM patients exhibited a more intense inflammatory status than PreDM patients, with significantly higher WBC values (8.70 vs. 7.84, *p* = 0.036) due to increased lymphocyte (LYM) counts (2.70 vs. 2.33, *p* = 0.035).

In the current study, we observed that patients with newly diagnosed T2DM experienced more severe anemia (*p* = 0.005) and had lower MCV levels (*p* = 0.046) compared to PreDM patients. When comparing known inflammatory indices between the newly diagnosed T2DM and PreDM patients, statistically significant lower index values were found for the PLR (89.10 vs. 113.40, *p* = 0.047), AISI (219.70 vs. 246.30, *p* = 0.038), and SII (377.90 vs. 411.80, *p* = 0.017).

Concerning the newly introduced MCVL and IIC indices, the MCVL index approached significance (*p* = 0.054), showing significantly lower values in newly diagnosed T2DM compared to those in PreDM.

PreDM and newly diagnosed T2DM patients showed statistically significant differences in the prognostic nutritional index (PNI) based on the number of lymphocytes (LYMs) in peripheral blood and serum albumin (ALB). Furthermore, patients who were newly diagnosed with T2DM exhibited moderate-to-severe malnutrition.

### 2.4. Comparing the Inflammatory Biomarkers Between the BMI Categories in the Studied Groups

Our study showed some particularities regarding comparing the inflammatory biomarkers between groups based on the BMI categories (Table 4).

In the PreDM group, after comparing the values between groups based on BMI categories, statistically significant variations were observed in PTX3 (one-way ANOVA test, *p* = 0.050), TNF-α (Kruskal–Wallis test, *p* = 0.0001), and IL-6 (Kruskal–Wallis test, *p* = 0.002). Serum levels showed a trend that correlated with increased BMI values, with the most significant changes seen in obese PreDM patients.

On the other hand, the one-way ANOVA test highlighted that, in the case of the hs-CRP, the differences between the BMI categories reached the significance limits (*p* = 0.056).

Within the T2DM cohort, we identified significant differences across the BMI categories with respect to PTX3, as determined by a one-way ANOVA test (*p* = 0.038), and TNF-α, assessed through the Kruskal–Wallis test (*p* = 0.028). Furthermore, the serum levels exhibited a trend positively correlating with the increasing BMI values, with the most pronounced alterations noted in patients with obesity and T2DM.

Comparing the known inflammatory indices between the PreDM and T2DM groups, we found statistically significant differences in values across BMI categories for only the AISI (Kruskal–Wallis test, *p* = 0.036 and *p* = 0.043, respectively). The index values exhibited an inverse trend, proportional to the increasing BMI values, with the lowest values observed in patients with obesity.

Regarding the newly introduced indices, our study revealed that, in the T2DM group, MCVL index values exhibited statistically significant differences among the BMI categories (one-way ANOVA test, *p* = 0.046). Furthermore, the index values showed an inverse trend that was proportional to the increasing BMI, with the lowest values observed in obese T2DM patients.

### 2.5. Comparing the Inflammatory Biomarker Values Between the HbA1c Quartiles in the Studied Groups

As shown in Table 5 and Figure 1, when we compared the PreDM group’s levels of inflammatory biomarkers to the HbA1c quartiles, we found statistically significant variations in PTX3 values between the Q1 + Q2 and Q3 + Q4 groups (*p* = 0.039). The serum levels exhibited a trend corresponding to the increased HbA1c values, with the most notable changes occurring in the Q3 + Q4 group.

In the case of TNF-α, hs-CRP, and IL-6, although higher mean levels were observed in the Q3 + Q4 group, these levels were not significantly higher when compared with the mean levels of the Q1 + Q2 group.

Comparing the index values of the already known indices, our study showed that the lower AISI index values were associated with HbA1C values above 5.46, and there were statistically significant differences between the Q1 + Q2 and Q3 + Q4 groups (*p* = 0.029). According to our analysis, the NLR, MLR, PLR, SII, MCVL, and IIC values for the HbA1c quartiles (Q1 + Q2 vs. Q3 + Q4) did not differ significantly.

Using the one-way ANOVA test, we found that T2DM patients (Table 6 and Figure 1) had statistically significant differences in values among the HbA1c quartiles concerning PTX3 levels (*p* = 0.048). The serum levels exhibited a trend corresponding to the increased values of HbA1c, with the most significant changes noted in the Q4 group (3429 pg/mL). Conversely, the Kruskal–Wallis test indicated similar trends for TNF-α levels, showing statistically significant differences across HbA1c quartiles (Kruskal–Wallis test, *p* <0.0001), where the higher TNF-α levels (399.20 pg/mL) were linked to HbA1c values exceeding 10.98.

Regarding the known indices, we found statistically significant differences in index values among the HbA1c quartiles for the PLR (one-way ANOVA test, *p* = 0.013) and AISI (Kruskal–Wallis test, *p* = 0.020). The index values exhibited an inverse trend correlated with increasing HbA1c levels, with the lowest index values noted in the Q4 group (78.37 and 195.80, respectively).

Regarding the newly introduced indices, a statistically significantly lower value was found among the IIC in the Q4 group (2.03), while the MCVL index reached the significance limits (one-way ANOVA test, *p* = 0.056).

### 2.6. Correlations Between Inflammatory Biomarkers and the Different Obesity-Related Indices in the PreDM Group

In the PreDM group (Figure 2), our study demonstrated that PTX3 levels exhibited a statistically significant and positive correlation with various parameters. Specifically, there was a moderate correlation with BMI (rho = 0.335, *p*-value = 0.040), and weak correlations were observed with the CKD-EPI (rho = 0.194, *p*-value = 0.033), weight (rho = 0.199, *p*-value = 0.029), and WHR (rho = 0.226, *p*-value = 0.023).

The hs-CRP and TNF-α levels correlated significantly better with the hematological and obesity-related indices, respectively. A strongly significant positive correlation was observed between hs-CRP and FPG (rho = 0.624, *p*-value = 0.0001) as well as TNF-α (rho = 0.454, *p*-value = 0.012) levels.

The hs-CRP levels exhibited positive moderate correlations with new hematological indices, specifically MCVL (rho = 0.361, *p*-value = 0.050) and IIC (rho = 0.290, *p*-value = 0.021). These levels also correlated positively with established hematological markers, such as NLR (rho = 0.372, *p*-value = 0.043), PLR (rho = 0.299, *p*-value = 0.019) and SII (rho = 0.262, *p*-value = 0.012). Additionally, weak to moderate correlations were observed with HbA1c (rho = 0.234, *p*-value = 0.013), height (rho = 0.286, *p*-value = 0.026), and WC values (rho = 0.225, *p*-value = 0.032).

Additionally, significantly moderate positive correlations were found between the TNF-α values and IL-6 (rho = 0.308, *p*-value = 0.047), AISI (rho = 0.436, *p*-value = 0.010), SII (rho = 0.404, *p*-value = 0.018), PLR (rho = 0.286, *p*-value = 0.011), NLR (rho = 0.261, *p*-value = 0.037), IIC (rho = 0.250, *p*-value = 0.022), HbA1c (rho = 0.396, *p*-value = 0.030), and height (rho = 0.376, *p*-value = 0.040).

Furthermore, the PNI values expressed a negative weak correlation with PTX3 (rho = −0.206, *p*-value = 0.042).

### 2.7. Correlations Between Inflammatory Biomarkers and the Different Obesity-Related Indices in the T2DM Group

Spearman’s correlation analysis showed that, in the T2DM group, the PTX3 levels correlated more strongly with new hematological indices (IIC, rho = 0.445, *p*-value = 0.014; MCVL, rho = 0.338, *p*-value = 0.048) and established hematological markers (NLR, rho = 0.367, *p*-value = 0.050; MLR, rho = 0.382, *p*-value = 0.037; AISI, rho = 0.279, *p*-value = 0.035) (Figure 3).

Moreover, the hs-CRP levels showed moderate-to-weak correlations with IIC (rho = 0.277, *p*-value = 0.013), MCVL (rho = 0.195, *p*-value = 0.023), SII (rho = 0.337, *p*-value = 0.049), NLR (rho = 0.346, *p*-value = 0.041), PLR (rho = 0.277, *p*-value = 0.019), and AISI (rho = 0.245, *p*-value = 0.033). Additionally, in contrast to PTX3, hs-CRP levels exhibited weak but statistically significant correlations with BMI (rho = 0.188, *p*-value = 0.028), weight (rho = 0.165, *p*-value = 0.035), WC (rho = 0.176, *p*-value = 0.031), and WHtR (rho = 0.140, *p*-value = 0.043).

Our study identified significant moderate-to-weak positive correlations between TNF-α levels and various obesity-related indices: WHtR (rho = 0.340, *p*-value = 0.046), WC (rho = 0.309, *p*-value = 0.027), WHR (rho = 0.279, *p*-value = 0.035), BAI (rho = 0.250, *p*-value = 0.018), and HC (rho = 0.194, *p*-value = 0.035).

Additionally, as in the case of the preDM group, the PNI values expressed a negative weak correlation with PTX3 (rho = −0.178, *p*-value = 0.037) levels.

Regarding the already known indices, we obtained statistically significant correlations between PLR and HC (rho = 0.383, *p*-value = 0.037), WHtR (rho = 0.404, *p*-value = 0.027), and BAI (rho = 0.555, *p*-value = 0.001) values. Also, BAI values correlated moderately with AISI (rho = 0.417, *p*-value = 0.022) and SII (rho = 0.436, *p*-value = 0.016).

### 2.8. Diagnostic Accuracy of the Biomarkers

Our study aimed to evaluate, through the analysis of the ROC curve, whether the investigated parameters (PTX3, hs-CRP, TNF-α, IL-6, NLR, MLR, PLR, AISI, SII, MCVL, and IIC) can distinguish the inflammatory status in PreDM and T2DM patients. For each parameter, a cut-off value was determined by maximizing the sum of sensitivity and specificity. Table 7 and Figure 4A–K present the ROC curves for the analyzed parameters.

In analyzing the area under the ROC curve (AUC) for the evaluated parameters, we observed that the most accurate diagnosis of the inflammatory status was achieved with IL-6 (86.60%) and PTX3 (72.00%), followed by MCVL (67.70%) and TNF-α (67.10%). Our study revealed a much lower diagnostic accuracy for hs-CRP and PLR, at 63.50% and 61.60%, respectively.

We selected cut-off levels for each of the examined inflammatory markers based on the sensitivity and specificity of the ROC curves in identifying the inflammatory status.

IL-6 had the best sensitivity and good specificity with 100.00% sensitivity and 60.00% specificity; the cut-off value was determined to be 40.30 pg/mL; the Youden index was 0.600.

PTX3 exhibited the highest specificity of 73.30% (with a sensitivity of 67.60%, a Youden index of 0.409 and a cut-off value of 1888.00 pg/mL). In our study, we found that, among the newly analyzed inflammatory markers, MCVL had the second highest specificity (66.70%) for identifying inflammatory status (with a sensitivity of 63.30%, a Youden index of 0.300, and a cut-off value determined to be 39.60).

Our research showed that while TNF-α and hs-CRP have a good AUC, their specificity was lower (56.70%).

In our study, we found the following results for various hematological indices: PLR demonstrated a sensitivity of 73.30% and a specificity of 60.00% (cut-off value of 101); SII exhibited a sensitivity of 56.70% and a specificity of 60.00% (cut-off value of 397); NLR showed a sensitivity of 65.00% and a specificity of 60.00% (cut-off value of 1.98); MLR had a sensitivity of 53.30% and a specificity of 56.70% (cut-off value of 0.212); IIC revealed a sensitivity of 61.70% and a specificity of 60.00% (cut-off value of 2.35); and AISI demonstrated a sensitivity of 56.70% and a specificity of 53.30% (cut-off value of 233).

## 3. Discussion

Diabetes mellitus notably raises the risk of cardiovascular disease, with patients being two to three times more susceptible compared to the general population. This makes cardiovascular complications the leading cause of morbidity and mortality in diabetic individuals. Therefore, effective diabetes management, including lifestyle changes and medications, is crucial to reduce cardiovascular disease incidence and improve patient outcomes [46].

Vascular changes in T2DM are associated with a subclinical inflammatory state, insulin resistance, and metabolic dysfunction. Studies show that T2DM correlates with elevated levels of acute-phase biomarkers like hs-CRP, which are linked to inflammation and the innate immune response. However, these elevated markers may also indicate other conditions, highlighting the need for more specific T2DM-related markers [41,43,47].

The pathophysiology of vascular changes is influenced by inflammatory markers, such as CRP and PTX3, which also serve as predictive indicators for cardiovascular events [48,49,50,51,52]. PTX3 is a member of the long pentraxin family of soluble proteins, activated by inflammatory stimuli. It plays a key role in localized inflammation and is important for innate immunity, especially in cardiovascular and renal diseases [52]. High plasma PTX3 levels are linked to an increased risk of cardiovascular diseases, according to several clinical studies [48,53].

In the present study, PTX3 and hs-CRP levels were higher in T2DM patients compared to PreDM patients, indicating a strong correlation between systemic inflammation and T2DM. To our knowledge, no studies have investigated the relationship between T2DM and various obesity-related indices in PreDM and newly diagnosed T2DM patients. Thus, based on the results obtained, our study provides new insights from a scientific perspective.

Our study elucidated that PTX3 values correspond with BMI categories, revealing serum levels that exhibit a trend proportional to increasing BMI values, with the highest concentrations noted in obese patients. These findings were consistent across both PreDM and newly diagnosed T2DM patients. We documented a 1.69-fold increase in PTX3 values for patients suffering from obese T2DM in comparison to those with obese PreDM. Moreover, when comparing the PTX3 values observed in overweight T2DM patients to those obtained from their overweight PreDM counterparts, we discerned a 1.58-fold increase.

Our study shows a strong link between PTX3 values and obesity, suggesting PTX3 as a potential biomarker for early diabetes diagnosis. The literature indicates that PTX3 may modulate atherosclerosis and is associated with diabetic micro- and macrovascular complications [40,41,42,43,44,45,50,52]. Following Dawood et al. [41], this study suggests that PTX3 may act as a specific diagnostic and prognostic biomarker for DN before the onset of overt CKD. Furthermore, it can differentiate between various stages of DN, while other routine tests, such as hs-CRP, are non-specific and unable to distinguish the stages of DN. Additionally, a study by Yang et al. [43] demonstrated that plasma PTX3 is closely linked to the development and progression of diabetic retinopathy (DR) and may serve as a more accurate predictor for DR onset than hsCRP in patients with diabetes mellitus (DM). Moreover, the varied alterations in PTX3 and adropin, as inflammatory markers, indicate their potential to predict renal injury in diabetic patients to differing extents and their association with the development of diabetic kidney disease [42].

In contrast, our study identified that, within the PreDM group, PTX3 values were found to be statistically significant and positively correlated with BMI and various obesity-related indices, including weight and WHR. Additionally, the hs-CRP levels demonstrated moderate-to-weak positive correlations with FGP, HbA1c, height, and WC values. Moreover, within the T2DM group, hs-CRP values exhibited weak yet statistically significant correlations with BMI, weight, WC, and WHtR, a distinction not observed with PTX3.

Additionally, supporting the earlier hypothesis, ROC curve analysis showed that PTX3 achieved the highest specificity of 73.30% (with a sensitivity of 67.60% and a Youden index of 0.409). For a cut-off value of 1888.00 pg/mL, it demonstrates a diagnostic accuracy of 72.00%. In the case of hs-CRP, our study showed significantly lower diagnostic accuracy (63.50%, specificity of 56.70%, and a cut-off value of 1025.00 pg/mL). These results align with Yang et al. [43], whose study indicated that ROC curves for developing DR using log PTX3 and log hs-CRP values revealed an AUC of 0.721 (with a cut-off value of 1406.0 pg/mL; sensitivity and specificity for DR development were 53.3% and 91.7%, respectively) and 0.614 (with a cut-off value of 841.7 pg/mL; the diagnostic sensitivity and specificity for DR development were 51.1% and 70.8%, respectively).

Our study observed a weak, negative, yet statistically significant correlation between PTX3 expression and PNI values. While this correlation is weak, PNI remains a key marker for assessing inflammation and nutrition in T2DM patients. Previously, we found relationships between PNI, the Glasgow Prognostic Score (GPS), and various obesity-related indices in individuals with T2DM or PreDM, suggesting these factors are independent predictors relevant to the American Diabetes Association (ADA) four pillars of diabetes management—glucose, blood pressure, lipids, and weight control [54].

Inflammation plays a significant role in the onset of T2DM and related metabolic diseases. Key risk factors for diabetic microvascular complications include the duration of diabetes, hyperglycemia, and insulin resistance. Additionally, chronic inflammation accelerates these complications, with both local and systemic inflammatory responses activated in the diabetic environment, contributing to the activation of many inflammatory cells [55,56,57,58,59,60,61,62].

The immune system components, including neutrophils, lymphocytes, and platelets, are essential for regulating immunity and are linked to microvascular complications in diabetes. Hematological indices offer better insights into immune responses compared to traditional indicators. However, research on indices like SII, AISI, MLR, PLR, and NLR as predictors of microvascular issues is inconsistent. For instance, Zhang et al. found no link between NLR and DN, while another study indicated a connection between NLR and DN, but not DR. Another study has associated both DR and diabetic peripheral neuropathy (DPN) with NLR and PLR [63,64,65,66,67]. A study by Li et al. [27] indicated that NLR was linked to the risk of DN, DR, and DPN, while only elevated levels of SII and PLR were related to the risk of DN and DR. Duan et al. [68] and Guo et al. [69] identified high SII and PLR lev-els as risk factors for diabetic nephropathy (DN). Research shows that patients with diabetic microvascular complications generally have higher neutrophil counts, while platelet and lymphocyte counts may not always change significantly. This may lead to an unstable relationship among diabetic microvascular complications, SII, and PLR [70,71].

Currently, no studies have investigated the relationship between established hemato-logical markers (NLR, MLR, PLR, AISI, SII) and emerging markers (MCVL, IIC) regarding obesity indices in PreDM patients and those newly diagnosed with T2DM.

In contrast to these hematological indices, including NLR, PLR, MLR, SII, and AISI, we aimed to assess the MCVL and IIC indices as independent predictive factors for differentiating inflammatory status in patients with and without T2DM. We proposed to test the two novel hematological indices in PreDM and newly diagnosed T2DM patients based on the results of the previous study [34,35,36,37]. When analyzing complete blood cell counts, our study highlighted that newly diagnosed T2DM patients exhibited more severe anemia than those with PreDM.

The measurement of the average size of erythrocytes, known as mean corpuscular volume (MCV), is strongly associated with erythrocyte diseases. It tends to be lower in diabetics, likely due to the higher incidence of iron deficiency anemia [72,73]. Hematological markers linked to hyperglycemia can be altered by diabetes. Prior research found that MCV was lower in DM patients than in non-DM patients [73,74]. The MCV represents the average volume of an RBC. It serves not only as an indicator of anemia but also as a marker of chronic inflammation. Research has shown it to be associated with heart failure, diabetes, vascular accidents, and venous thromboembolism [75].

The MCVL index, calculated by dividing the absolute number of erythrocytes MCV by the absolute number of lymphocytes, showed significantly lower values in newly diagnosed T2DM patients. There were significant differences across BMI categories, with index values demonstrating an inverse trend proportional to BMI increases, yielding the lowest values in obese T2DM patients. Additionally, the MCVL index displayed an inverse trend proportional to rising HbA1c levels, with the lowest index values observed in the Q4 T2DM group. Furthermore, the IIC, obtained by multiplying the MCV by the erythrocyte distribution width (RDW) and the absolute number of neutrophils, and then dividing by the absolute number of lymphocytes multiplied by 1000, indicated a statistically significantly lower value in the Q4 T2DM group. Moreover, Spearman’s analysis illustrated that, within the T2DM group, PTX3 values correlated much better with new hematological indices (IIC, rho = 0.445, *p*-value = 0.014; MCVL, rho = 0.338, *p*-value = 0.048). Additionally, hs-CRP values exhibited moderate-to-weak correlations with IIC and MCVL in both groups. Our study found that MCVL had a specificity of 66.70% in identifying the inflammatory status in patients with and without T2DM (sensitivity of 63.30%, Youden index 0.300, and a cut-off value of 39.60). IIC demonstrated a sensitivity of 61.70% and a specificity of 60.00% (cut-off value of 2.35).

We recognize that our study has inherent limitations. Furthermore, we did not compute the sample size simulation because we lacked pilot research and previous data cited in the literature, which prevented us from determining the effect size. First, only 90 patients were included in the sample, a modest number. This restricts the generalizability of the findings and could lower the study’s statistical power, especially when examining subgroups like BMI categories and HbA1c quartiles. Another limitation is that the study is a retrospective design with selection bias (such as excluding patients with microvascular complications), which may limit the generalizability of the results. Thus, further research should investigate how obesity-related indices relate to pentraxins and hematological indices across various patient groups with diabetes and prediabetes. Furthermore, the investigation took place in two university clinical hospitals representative of Dolj County, which may have introduced geographic and demographic biases. Lastly, the observational design of the study limits our ability to draw causal conclusions regarding the relationships between hematological indices, pentraxins, and obesity-related indices.

## 4. Materials and Methods

We conducted a six-month epidemiological, non-interventional, retrospective study. This study enrolled one hundred and ninety consecutive patients with newly diagnosed T2DM. In comparison, the control group consisted of thirty patients with PreDM who matched the inclusion criteria for age, gender ratio, and urban/rural residence.

The study was conducted under the Declaration of Helsinki and approved by the Ethics Committee of the Clinical Municipal Hospital Filantropia (no. 886/15 January 2024) and the Emergency County Clinical Hospital of Craiova (no. 2371/14 January 2022), Dolj, Romania.

### 4.1. Patient Selection

To be eligible for the study, participants needed to have T2DM or PreDM and be over eighteen. They were chosen from the Diabetes, Nutrition, and Metabolic Diseases Outpatient Departments at the Clinical Municipal Hospital Filantropia and the Emergency County Clinical Hospital of Craiova. After signing an informed consent form, each subject willingly agreed to participate in the study.

The study did not include patients with diabetic peripheral polyneuropathy, diabetic kidney disease, or diabetic retinopathy, which are chronic microvascular complications of T2DM. It also excluded patients with cancer, those under 18 years of age, pregnant women, individuals with type 1 diabetes, those who had an acute infection or inflammatory condition in the previous month, and people with chronic infections or inflammatory diseases.

Sixty out of one hundred and ninety T2DM patients completed the study and were included in the final analysis, while one hundred and thirty were lost: diabetic peripheral polyneuropathy (*n* = 40), diabetic kidney disease (*n* = 50), and diabetic retinopathy (*n* = 40).

### 4.2. Diagnosis of Diabetes and Prediabetes in Laboratories

PreDM may be defined according to one of the following criteria: HbA1c levels ranging from 5.7% to less than 6.5%, FPG levels between 5.6 and 7.0 mmol/L, a 2hPG between 7.8 and 11.0 mmol/L, or a diagnosis rendered by a qualified medical professional. These represent the initial four criteria for identifying PreDM [76]. The PreDM group includes patients with hypertension, dyslipidemia (high triglycerides and/or low HDL cholesterol), and obesity (particularly visceral or abdominal obesity).

A diagnosis of diabetes is established if one or more of the following criteria are fulfilled: a medical diagnosis confirmed by the patient’s healthcare providers; a random blood glucose level of 11.1 mmol/L or higher; an HbA1c value exceeding 6.5%; an FPG level of 7.0 mmol/L or higher; a two-hour blood glucose level greater than 11.1 mmol/L following an oral glucose tolerance test (OGTT); or a random glucose measurement accompanied by symptoms of hyperglycemia (such as polyuria, polydipsia, and unexplained weight loss) or hyperglycemic crises.

### 4.3. Medical Background, Evaluation of Biometric Parameters, and Demographic Information

An interview questionnaire was designed to collect information on anthropometric measurements, medical factors, laboratory test findings, and demographic and lifestyle details.

The demographic variables included age, sex, monthly household income, and education level. Factors related to lifestyle and health comprised a family history of hypertension, diabetes mellitus, cardiovascular illnesses, alcohol consumption, smoking habits, and the time devoted to purposeful moderate physical activity each week. Regarding the patients’ medication, some patients were on treatment with antihypertensives (perindopril, perindopril/indapamide combination), and statins (atorvastatin, rosuvastatin).

Regarding the dietary factor, most patients did not have a diet established by a specialist. Also, following the consultation, it was noted that the subjects did not perform physical activity more than 4 days per week and were considered sedentary.

### 4.4. Evaluation of Different Obesity-Related Indices (BMI, WHR, WHtR, and BAI)

The BMI was calculated using the participants’ weight and height measurements. The calculation formula is BMI = weight (kilograms)/height^2^ (meters). The patient’s nutritional status was assessed according to BMI, following the WHO criteria guidelines [77]. BMI was categorized into normal weight (18.5–22.9 kg/m^2^), overweight (23.0–25.0 kg/m^2^), and obese (>25.0 kg/m^2^). A weight scale was used to measure weight, while height was measured with a measuring stick attached to the weight scale.

Measurements of the WC were taken at the midpoint between the lower border of the rib cage and the upper iliac crest, while the HC was measured over the femoral trochanters. Abdominal obesity was assessed using the WHR, calculated with the formula WC (cm)/HC (cm). The WHtR was also employed to evaluate visceral adiposity, calculated using the formula WC (cm)/height (m). To compute the body adiposity index (BAI), the formula used was ((hip circumference)/((height)^1.5)) − 18 [78]. We categorized WHR, WHtR, and BAI into quarters due to the lack of established categories.

### 4.5. Laboratory Investigations

After collecting anthropometric data, we brought the subjects to the laboratory for further investigation.

#### Sample Collection

Approximately 5 mL of venous blood was collected from patients in additive-free tubes (Becton Dickinson vacutainer, Franklin Lakes, NJ, USA) as part of the biological samples. Following standard procedure, the clot was separated within 4 h of collection by centrifugation (Hermle AG, Gosheim, Baden-Württemberg, Germany) at 3000× *g* for 10 min. The serum sample vials were labeled for each patient, stored at temperatures between −20 °C and −80 °C to facilitate longer processing times, and tightly sealed to prevent contamination. Freezing and thawing cycles were avoided, and frozen specimens were allowed to thaw at room temperature before processing patient samples.

Laboratory data, including BUN, CREA, uric acid (UA), FPG, 2hPG, HbA1c, TC, TG, LDL-C, HDL-C, CRP, and ALB, were determined using the chemiluminescence immunological technique and an automatic immunoassay analyzer (Cobas e411, Roche Diagnostics GmbH, Mannheim, Germany).

A complete blood count (CBC) was performed using peripheral venous blood collected in vacutainer tubes containing ethylene-diamine-tetra-acetic acid (EDTA) as an anticoagulant. Utilizing flow cytometry and Coulter’s principle, we obtained an extended leukocyte formula of 5 diff (Alinity Abbott, Abbott Park, IL, USA) and determined the hemoleucogram markers: hemoglobin (Hb), white blood cells/leukocytes (WBCs), neutrophils (NEUs), lymphocytes (LYMs), monocytes (MONs), platelets (PLTs), and hemoglobin (Hb). The inflammation indices derived from the blood cell count, NLR, MLR, PLR, AISI, SII, MCVL, and IIC were calculated based on these findings.

Measurements of serum creatinine were made, and the Chronic Kidney Disease Epidemiology Collaboration (CKD-EPI) formula [79] and the Modification of Diet in Renal Disease Study (MDRD-Study) [80] were used to determine the eGFR.

### 4.6. Immunological Assessment

The Enzyme-Linked Immunosorbent Assay (ELISA) method was used in the Immunology Laboratory of the University of Medicine and Pharmacy of Craiova to determine serum quantities of PTX3, hs-CRP, TNF-α, and IL-6.

For each of the following mediators, we used commercially available test sets: PTX3 (Catalog# E-EL-H6081; sensitivity: 4.12 pg/mL; detection range: 6.86–5000 pg/mL), hs-CRP (Catalog# E-EL-H5134; sensitivity: 9.38 pg/mL; detection range: 15.63–1000 pg/mL), from the Elabscience (Houston, TX, USA); TNF-α (Catalog# BMS223-4; sensitivity: 2.3 pg/mL; assay range: 7.8–500 pg/mL), IL-6 (Catalog# BMS213-2; sensitivity: 0.92 pg/mL; assay range: 1.56–100 pg/mL), from Invitrogen, Thermo Fisher Scientific, Inc. (Waltham, MA, USA).

Following the manufacturer’s instructions and recommended procedure, each sample was diluted after freezing. A standard optical analyzer with a wavelength of 450 nm was used for the procedure.

### 4.7. Calculations for the Prognostic Nutritional Index and Glasgow Prognostic Score

Based on the absolute lymphocyte count and serum ALB level, the PNI was calculated. The PNI was calculated according to the acknowledged formula: 10 × serum albumin (g/dL) + 0.5% × total lymphocyte number (per mm^3^) [81]. Interpretation: PNI value ≥ 50—Normal, PNI value < 50—Mild malnutrition, PNI value< 45—Moderate-to-severe malnutrition, PNI value< 40—Serious malnutrition.

CRP and ALB levels were used to calculate the GPS, and patients with CRP ≤ 10 mg/L and ALB ≥ 35 g/L were allocated to the GPS-0 group. Patients with only CRP > 10 mg/L were assigned to the GPS-1 group. Patients with CRP levels greater than 10 mg/L and ALB levels below 35 g/L were assigned a score of 2 [82].

### 4.8. Statistical Analysis

We processed and handled patient data from medical records using Microsoft Excel. To analyze the data, we utilized GraphPad Prism 10.4.2.633 Version (GraphPad Software, San Diego, CA, USA). The data were examined for normality using the Shapiro–Wilk and Kolmogorov–Smirnov tests.

The clinical and demographic characteristics of the study participants were compiled using descriptive statistics. The means and SD of the following variables are presented, all of which had normal distributions: PTX3, hs-CRP, WBC, NEU, LYM, MON, PLT, RDW, NLR, MLR, PLR, MCVL, IIC, BMI, PNI, height, weight, WC, HC, WHR, WHtR, and BAI. It was shown that the distributions of TNF-α, IL-6, RBC, MCV, AISI, and SII were non-normal, and the data are represented as the median with an interquartile range. Frequencies and percentages were used to represent categorical variables.

Continuous variables were evaluated using the Student’s *t*-test/One-Way ANOVA or the Mann–Whitney test/Kruskal–Wallis’s test (used for non-Gaussian distributions) to find the difference between groups, and the *χ*^2^ test was used for categorical variables.

To check for any significant correlations between the levels or values of the parameters, Spearman’s coefficients (−1 < rho < 1) were used.

ROC curve analysis was used to evaluate the diagnostic accuracy of the researched markers. We quantified performance using the AUC and p-statistics, comparing the calculated AUC against a threshold of AUC = 0.5, which indicates a weak discriminative marker. To identify the cut-off values that yielded maximum accuracy, we calculated the sensitivity, specificity, and Youden index (sensitivity + specificity − 1) for each marker at the different threshold values investigated.

## 5. Conclusions

This is the first study demonstrating an independent relationship between pentraxins (PTX3, hs-CRP) and various obesity-related indices in patients with PreDM and T2DM. Additionally, ROC analysis indicated that PTX3 and the newly introduced MCVL index accurately diagnosed the inflammatory status compared to IL-6. According to these results, PTX3 and the MCVL index can act as independent predictors for inflammatory status in patients with PreDM and T2DM. Furthermore, these parameters could serve as promising candidate biomarkers to assist in PreDM and newly diagnosed T2DM management. Our results can certainly provide a basis for future investigations and long-term, multicenter studies. These studies could further elucidate the dynamic relationship between changes in inflammatory biomarkers and the progression of diabetes, potentially leading to more tailored treatment approaches. By enhancing our understanding of these indicators, we may improve patient outcomes and streamline the management of diabetes.

## Figures and Tables

**Figure 1 ijms-26-03661-f001:**
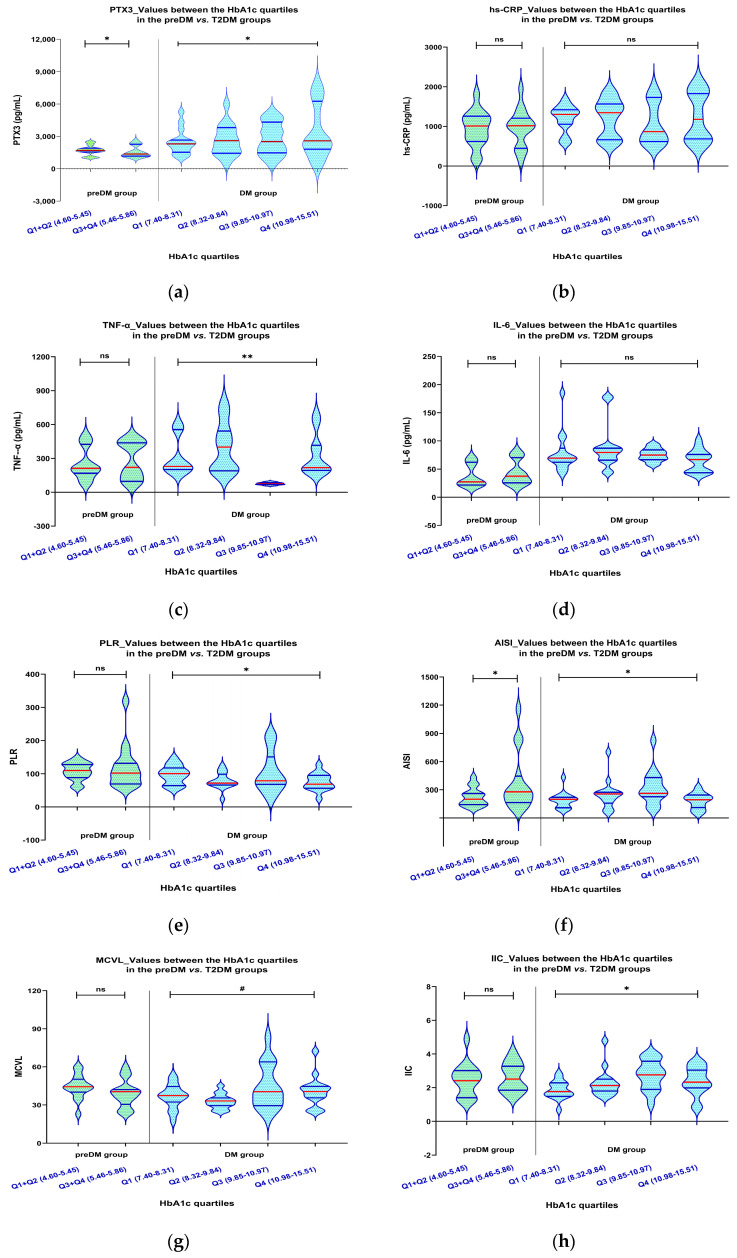
The PTX3 (**a**), hs-CRP (**b**), TNF–α (**c**), IL–6 (**d**), PLR (**e**), AISI (**f**), MCVL (**g**), and IIC (**h**) levels for patients with preDM and T2DM vary in the HbA1c quartiles. When we compared the PreDM group’s levels of inflammatory biomarkers to the HbA1c quartiles, we found statistically significant variations in PTX3 (*p* = 0.039), and AISI (*p* = 0.029) values between the Q1 + Q2 and Q3 + Q4 groups. Using the one-way ANOVA test, we found that T2DM patients had statistically significant differences in values among the HbA1c quartiles among PTX3 (*p* = 0.048), TNF–α (*p* < 0.0001), PLR (*p* = 0.013), AISI (*p* = 0.020), and IIC (*p* = 0.027); while the MCVL index reached the significance limits (one-way ANOVA test, *p* = 0.056). The violin plot represents values of the inflammatory biomarkers; horizontal red lines represent median values accompanied by the quartiles represented by horizontal blue lines. PTX3: pentraxin 3; hs–CRP: high–sensitivity C–reactive protein; TNF–α: tumor necrosis factor alpha; IL–6: interleukin 6; PLR: platelet-lymphocyte ratio; AISI: aggregate index of systemic inflammation; MCVL: ratio between the mean corpuscular volume/lymphocytes; IIC: cumulative inflammatory index; *, *p* ≤ 0.05; **, *p* < 0.0001; #, reached the significance limit; ns: statistically not significant differences.

**Figure 2 ijms-26-03661-f002:**
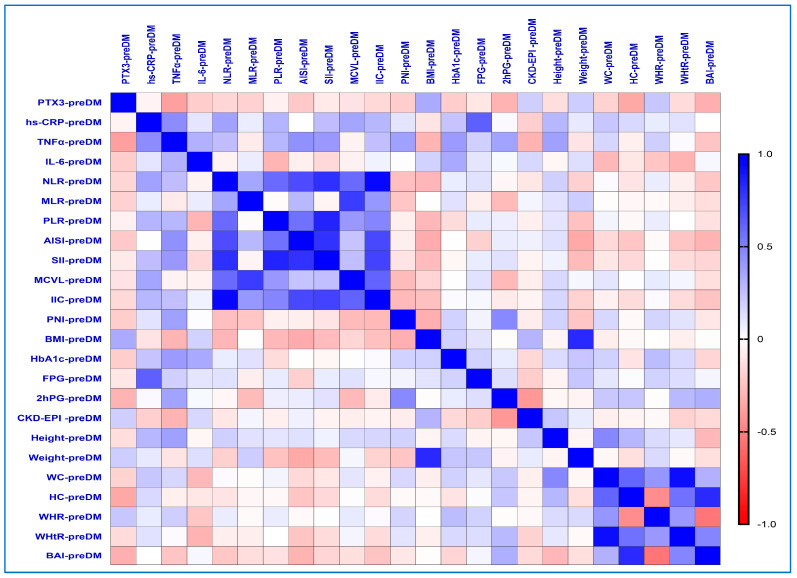
Correlations matrix between inflammatory biomarkers and the different obesity-related indices in the PreDM group. The correlation heatmap shows how the measured indicators relate to one another. Strong positive correlations are indicated by bright blue, whereas strong negative correlations are indicated by bright red.

**Figure 3 ijms-26-03661-f003:**
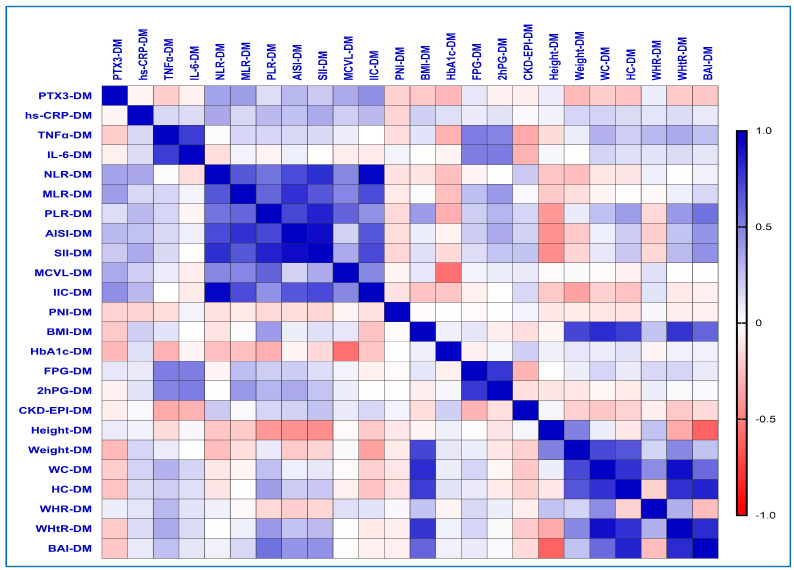
Correlations matrix between inflammatory biomarkers and the different obesity-related indices in the T2DM group. The correlation heatmap shows how the measured indicators relate to one another. Strong positive correlations are indicated by bright blue, whereas strong negative correlations are indicated by bright red.

**Figure 4 ijms-26-03661-f004:**
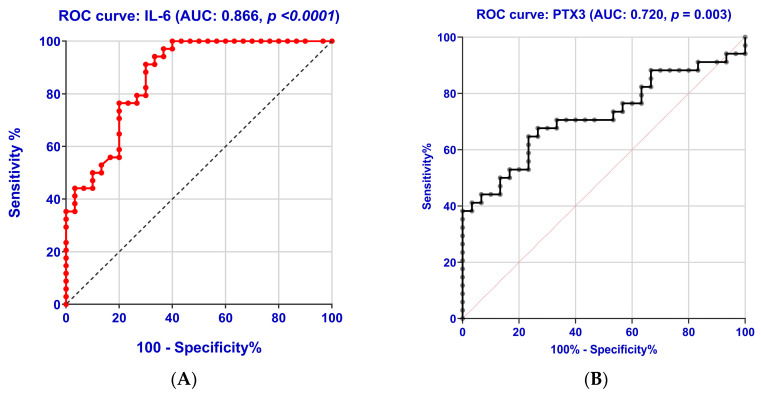
Receiver operating characteristic (ROC) curve for IL-6 (**A**), PTX3 (**B**), MCVL (**C**), TNF-α (**D**), hs-CRP (**E**), PLR (**F**), SII (**G**), NLR (**H**), MLR (**I**), IIC (**J**), and AISI (**K**). ROC analysis showed that the PTX3 (AUC was 0.720; *p* = 0.003; cut-off value 1888.00 pg/mL, with 67.60% sensitivity and the highest specificity of 73.30%) and MCVL index (AUC was 0.677; *p* = 0.047; cut-off value 39.60, with 63.30% sensitivity and the second highest specificity of 66.70%) had a good accurate diagnosis compared with IL-6 (AUC was 0.866; *p* < 0.0001; cut-off value 40.30 pg/mL, with 100.00% sensitivity and 60.00% specificity). IIC showed 61.70% sensitivity and 60.00% specificity, with an AUC of 0.572, *p* = 0.027, and a cut-off value of 2.35. In our study, we found the following results for various hematological indices: PLR demonstrated 73.30% sensitivity and 60.00% specificity (cut-off value of 101); SII exhibited 56.70% sensitivity and 60.00% specificity (cut-off value of 397); NLR showed a sensitivity of 65.00% and a specificity of 60.00% (cut-off value of 1.98); MLR had a sensitivity of 53.30% and a specificity of 56.70% (cut-off value of 0.212); while AISI demonstrated a sensitivity of 56.70% and a specificity of 53.30% (cut-off value of 233). PTX3: pentraxin 3; hs-CRP: high-sensitivity C-reactive protein; TNF-α: tumor necrosis factor alpha; IL-6: interleukin 6; NLR: neutrophil-lymphocyte ratio; MLR: monocyte-lymphocyte ratio; PLR: platelet-lymphocyte ratio; AISI: aggregate index of systemic inflammation; SII: systemic immune-inflammation index; MCVL: ratio between the mean corpuscular volume/lymphocytes; IIC: cumulative inflammatory index.

**Table 1 ijms-26-03661-t001:** Demographic parameters of the patients stratified according to the groups under study.

Variables		PreDM (*n* = 30)	T2DM (*n* = 60)	*p*-Value from Pearson’s Chi-Squared/ Student’s *t*-Test
Age (years)	Mean	49.87	53.43	0.013 *
	±SD	7.68	5.46	
Gender, *n* (%)	Male	13 (43.33%)	28 (46.67%)	0.765
	Female	17 (56.67%)	32 (53.33%)	
Residence, *n* (%)	Urban	12 (40.00%)	24 (40.00%)	0.819
	Rural	18 (60.00%)	36 (60.00%)	
Smoking habit, *n* (%)		14 (46.67%)	32 (53.33%)	0.553
Drinking habit, *n* (%)		10 (33.33%)	24 (40.00%)	0.541
Hypertension, *n* (%)		23 (76.67%)	45 (75.00%)	0.863
Dyslipidemia, *n* (%)		21 (70.00%)	48 (80.00%)	0.293
Hepatosteatosis, *n* (%)		16 (53.33%)	38 (63.33%)	0.364
SBP, mmHg	Mean	131.70	135.6	0.271
	±SD	16.84	15.18	
DBP, mmHg	Mean	78.90	83.82	0.044 *
	±SD	11.71	10.25	
BMI, kg/m^2^	Mean	30.75	31.38	0.644
	±SD	6.99	5.57	
BMI category, *n* (%)				
Normal (18.5–24.9 kg/m^2^)		9 (30.00%)	12 (20.00%)	0.409
Overweight (25–29.9 kg/m^2^)		11 (36.67%)	14 (23.33%)	0.325
Obese (≥30 kg/m^2^)		10 (33.33%)	34 (56.67%)	0.207
Height, m	Mean	1.61	1.70	0.0001 *
	±SD	0.08	0.09	
Weight, kg	Mean	84.84	87.11	0.557
	±SD	18.98	16.21	
WC, cm	Mean	97.99	104.70	0.026 *
	±SD	13.82	12.80	
HC, cm	Mean	107.10	108.90	0.513
	±SD	11.83	14.46	
WHR	Mean	0.91	0.98	0.006 *
	±SD	0.16	0.08	
WHtR	Mean	0.60	0.62	0.488
	±SD	0.08	0.09	
BAI	Mean	30.72	35.05	0.007 *
	±SD	7.30	7.16	

SBP: systolic blood pressure; DBP: diastolic blood pressure; BMI: body mass index; WC: Waist circumference; HC: Hip circumference; WHR: waist to hip ratio; WHtR: waist to height ratio; BAI: body adiposity index; SD: standard deviation; * *p* < 0.05: statistically significant.

**Table 2 ijms-26-03661-t002:** Comparison between the studied groups as regards laboratory parameters.

Variables		PreDM (*n* = 30)	T2DM (*n* = 60)	*p*-Value from Student’s *t*-Test/ Mann–Whitney Test
FPG (mg/dL)	Median	108.0	212.0	<0.0001 *
	range	100.0–122.0	123.0–247.0	
2hPG (mg/dL)	Mean	164.40	331.60	<0.0001 *
	±SD	14.69	41.18	
HbA1c (%)	Median	5.45	9.85	<0.0001 *
	range	4.60–5.86	7.40–15.51	
TC (mg/dL)	Mean	185.0	221.10	0.002 *
	±SD	52.26	49.27	
TG (mg/dL)	Mean	134.8	187.4	0.016 *
	±SD	71.0	106.1	
LDL-C (mg/dL)	Mean	102.5	137.2	0.001 *
	±SD	47.13	43.94	
HDL-C (mg/dL)	Mean	52.98	44.99	0.010 *
	±SD	13.23	13.60	
eGFR (CKD-EPI) (mL/min/1.73 m^2^)	Mean	86.23	85.85	0.929
	±SD	18.21	19.79	
BUN (mg/dL)	Mean	37.33	41.18	0.255
	±SD	14.80	15.79	
Crea (mg/dL)	Median	0.74	0.80	0.023 *
	range	0.47–1.53	0.56–1.67	
UA (mg/dL)	Mean	5.01	5.17	0.657
	±SD	1.37	1.74	
ALB (g/dL)	Mean	6.17	3.87	<0.0001 *
	±SD	0.34	0.81	

FPG: fasting plasma glucose; 2hPG: two-hour plasma glucose after a 75 g oral glucose tolerance test; HbA1c: glycosylated hemoglobin A1c; TC: total cholesterol; TG: total triglycerides; LDL-C: low-density lipoprotein cholesterol; HDL-C: high-density lipoprotein cholesterol; e-GFR: estimated glomerular filtration rate; CKD-EPI: chronic kidney disease epidemiology collaboration; BUN: blood urea nitrogen; Crea: creatinine; UA: uric acid; ALB: albumin; SD: standard deviation; * *p* < 0.05: statistically significant.

**Table 3 ijms-26-03661-t003:** Comparison between the studied groups as regards inflammatory biomarkers.

Variables		PreDM (*n* = 30)	T2DM (*n* = 60)	*p*-Value from Student’s *t*-Test/ Mann–Whitney Test
PTX3 (pg/mL)	Mean	1649.00	2826.00	0.0009 *
	±SD	494.30	1795.00	
hs-CRP (pg/mL)	Mean	954.20	1193.00	0.048 *
	±SD	453.30	495.00	
TNF-α (pg/mL)	Median	84.23	194.00	0.019 *
	range	30.77–203.90	164.70–278.10	
IL-6 (pg/mL)	Median	29.96	72.34	<0.0001 *
	range	17.52–82.41	40.78–185.20	
ESR (mm/1st h)	Mean	24.27	39.03	0.0004 *
	±SD	14.63	19.18	
RBC (×10^3^/μL)	Median	4.85	4.46	0.005 *
	range	3.43–6.73	1.39–5.36	
WBC (×10^3^/μL)	Mean	7.84	8.70	0.036 *
	±SD	1.87	1.76	
NEU (×10^3^/μL)	Mean	4.73	5.17	0.759
	±SD	1.43	2.87	
LYM (×10^3^/μL)	Mean	2.33	2.70	0.035 *
	±SD	0.71	0.78	
MON (×10^3^/μL)	Mean	0.53	0.57	0.415
	±SD	0.16	0.19	
PLT (×10^3^/μL)	Mean	225.70	243.70	0.254
	±SD	72.45	65.09	
MCV (fL)	Median	96.19	90.79	0.046 *
	range	78.60–118.00	64.50–98.40	
RDW (%)	Mean	13.23	12.71	0.154
	±SD	1.17	0.99	
NLR	Mean	2.15	1.90	0.120
	±SD	0.78	0.67	
MLR	Mean	0.24	0.22	0.689
	±SD	0.08	0.12	
PLR	Mean	113.40	89.10	0.047 *
	±SD	49.97	39.27	
AISI	Median	246.30	219.70	0.038 *
	range	93.81–1160.00	34.82–823.20	
SII	Median	411.80	377.90	0.017 *
	range	240.30–1657.00	54.40–1066.00	
MCVL	Mean	41.70	38.69	0.054 **
	±SD	10.41	12.49	
IIC	Mean	2.54	2.32	0.249
	±SD	0.91	0.84	
PNI	Mean	61.99	38.70	<0.0001 *
	±SD	3.44	8.09	

PTX3: pentraxin 3; hs-CRP: high-sensitivity C-reactive protein; TNF-α: tumor necrosis factor-alpha; IL-6: interleukin 6; ESR: erythrocyte sedimentation rate; RBC: red blood cells; WBC: white blood cells/leukocytes; NEU: neutrophils; LYM: lymphocytes; MON: monocytes; PLT: platelets; MCV: the mean corpuscular volume; RDW: erythrocyte distribution width; NLR: neutrophil-lymphocyte ratio; MLR: monocyte-lymphocyte ratio; PLR: platelet-lymphocyte ratio; AISI: aggregate index of systemic inflammation; SII: systemic immune-inflammation index; MCVL: ratio between the mean corpuscular volume/lymphocytes; IIC: cumulative inflammatory index; PNI: prognostic nutritional index; SD: standard deviation; * *p* < 0.05: statistically significant; **: reached the significance limit.

**Table 4 ijms-26-03661-t004:** Evaluation of Parameters According to the BMI Categories in the Studied Groups.

Variables		PreDM				T2DM			
		Normal (*n* = 9)	Overweight (*n* = 11)	Obese (*n* = 10)	*p*-Value from One-Way ANOVA/ Kruskal–Wallis Test	Normal (*n* = 12)	Overweight (*n* = 14)	Obese (*n* = 34)	*p*-Value from One-Way ANOVA/ Kruskal–Wallis Test
PTX3 (pg/mL)	Mean	1318	1789	1792	0.050 *	2611	2826	3019	0.038 *
	±SD	320	425.3	568.4		2033	1795	1285	
hs-CRP (pg/mL)	Mean	656	1079	1085	0.056 **	1152	1179	1193	0.068
	±SD	378.9	363.8	497.7		483.5	532.9	495	
TNF-α (pg/mL)	Median	102.10	218.90	432.60	0.0001 *	207.90	320.80	329.10	0.028 *
	range	76.92–211.10	87.27–463.50	176.50–509.80		167.20–90.90	189.60–709.90	172.20–01.70	
IL-6 (pg/mL)	Median	25.46	29.76	69.91	0.002 *	71.82	72.34	75.93	0.463
	range	19.24–62.03	19.24–54.56	17.52–82.41		41.92–185.20	40.78–87.04	44.22–110.80	
NLR	Mean	2.10	2.03	1.94	0.837	2.32	2.12	1.99	0.665
	±SD	0.92	0.55	0.72		0.76	0.90	0.70	
MLR	Mean	0.21	0.23	0.27	0.280	0.26	0.25	0.24	0.942
	±SD	0.06	0.05	0.11		0.18	0.15	0.14	
PLR	Mean	113.8	120.7	105.2	0.788	110.2	94.0	89.1	0.308
	±SD	46.1	70.32	22.83		55.14	32.13	37.67	
AISI	Median	278.90	237.50	233.30	0.036 *	263.50	220.10	201.00	0.043 *
	range	145.5–834.6	93.8–1160.0	101.2–433.2		84.4–436.9	51.9–823.2	75.1–510.8	
SII	Median	460.90	398.40	391.40	0.097	421.30	394.10	387.10	0.071
	range	316.2–866.3	246.6–1284.0	240.30–1657.0		165.5–769.0	121.6–1066.0	83.6–1064.0	
MCVL	Mean	42.80	41.30	40.90	0.916	44.60	40.00	37.10	0.046 *
	±SD	14.40	8.10	9.50		18.70	13.90	13.50	
ICC	Mean	2.71	2.47	2.45	0.790	2.42	2.40	2.22	0.756
	±SD	0.79	0.99	0.98		1.07	0.85	0.71	
PNI	Mean	63.22	61.76	59.90	0.108	41.90	38.33	37.51	0.232
	±SD	3.724	2.790	3.383		9.04	9.05	7.19	

PTX3: pentraxin 3; hs-CRP: high-sensitivity C-reactive protein; TNF-α: tumor necrosis factor-alpha; IL-6: interleukin 6; NLR: neutrophil-lymphocyte ratio; MLR: monocyte-lymphocyte ratio; PLR: platelet-lymphocyte ratio; AISI: aggregate index of systemic inflammation; SII: systemic immune-inflammation index; MCVL: ratio between the mean corpuscular volume/lymphocytes; IIC: cumulative inflammatory index; PNI: prognostic nutritional index; SD: standard deviation; * *p* < 0.05: statistically significant; **: reached the significance limit.

**Table 5 ijms-26-03661-t005:** Comparing the Inflammatory Biomarkers Values Between the HbA1c Quartiles in the PreDM Group.

Variables		Quartiles of HbA1c		
		Q1 + Q2 (4.60–5.45) (*n* = 15)	Q3 + Q4 (5.46–5.86) (*n* = 15)	*p*-Value from Student’s *t*-Test/ Mann–Whitney Test
PTX3 (pg/mL)	Mean	1576	1722	0.039 *
	±SD	536.1	455.3	
hs-CRP (pg/mL)	Mean	944.4	963.9	0.909
	±SD	449.2	473.0	
TNF-α (pg/mL)	Median	210.0	218.9	0.967
	range	79.9–509.8	76.9–489.9	
IL-6 (pg/mL)	Median	27.32	37.33	0.095
	range	19.2–71.4	17.5–82.4	
NLR	Mean	2.23	2.07	0.577
	±SD	0.82	0.76	
MLR	Mean	0.25	0.23	0.500
	±SD	0.11	0.05	
PLR	Mean	119.5	107.4	0.517
	±SD	88.4	26.2	
AISI	Median	278.9	201.2	0.029 *
	range	93.8–1160.0	104.7–433.2	
SII	Median	418.8	404.9	0.480
	range	240.3–866.3	246.3–1657.0	
MCVL	Mean	44.38	39.02	0.162
	±SD	9.54	10.87	
IIC	Mean	2.62	2.46	0.661
	±SD	0.86	0.98	
PNI	Mean	61.85	61.52	0.799
	±SD	3.79	3.18	

PTX3: pentraxin 3; hs-CRP: high-sensitivity C-reactive protein; TNF-α: tumor necrosis factor alpha; IL-6: interleukin 6; NLR: neutrophil-lymphocyte ratio; MLR: monocyte-lymphocyte ratio; PLR: platelet-lymphocyte ratio; AISI: aggregate index of systemic inflammation; SII: systemic immune-inflammation index; MCVL: ratio between the mean corpuscular volume/lymphocytes; IIC: cumulative inflammatory index; PNI: prognostic nutritional index; SD: standard deviation; * *p* < 0.05: statistically significant.

**Table 6 ijms-26-03661-t006:** Comparing the Inflammatory Biomarkers Values Between the HbA1c Quartiles in the T2DM Group.

Variables			Quartiles of HbA1c			
		Q1 (7.40–8.31)	Q2 (8.32–9.84)	Q3 (9.85–10.97)	Q4 (10.98–15.51)	*p*-Value from One-Way ANOVA/ Kruskal–Wallis Test
PTX3 (pg/mL)	Mean	2460	2639	2785	3429	0.048 *
	±SD	1175	1451	1494	2482	
hs-CRP (pg/mL)	Mean	1067	1212	1203	1241	0.782
	±SD	529	475	340	615	
TNF-α (pg/mL)	Median	75	217	227.8	399.2	<0.0001 *
	range	57.4–95.4	182–699	177.4–613.0	172.2–801.7	
IL-6 (pg/mL)	Median	66.85	69.44	75.00	79.63	0.211
	range	41.92–102.80	40.78–185.20	57.43–95.37	44.22–178.60	
NLR	Mean	2.28	2.04	1.96	1.81	0.159
	±SD	0.78	0.73	0.55	0.59	
MLR	Mean	0.33	0.25	0.22	0.21	0.148
	±SD	0.20	0.14	0.14	0.10	
PLR	Mean	107.1	106.8	85.47	78.37	0.013 *
	±SD	36.8	59.17	26.61	25.52	
AISI	Median	268.3	263.5	199.8	195.8	0.020 *
	range	51.9–702.4	115.5–823.2	84.4–432.6	53.5–331.2	
SII	Median	431.7	394.1	376.6	371.2	0.066
	range	116.6–1064.0	165.5–839.7	121.6–1066.0	83.7–714.7	
MCVL	Mean	44.92	40.27	39.64	36.08	0.056 **
	±SD	19.44	12.56	13.77	11.85	
IIC	Mean	2.74	2.43	2.29	2.03	0.027 *
	±SD	0.85	0.91	0.74	0.64	
PNI	Mean	39.43	39.60	38.93	36.84	0.782
	±SD	9.35	7.14	8.39	7.87	

PTX3: pentraxin 3; hs-CRP: high-sensitivity C-reactive protein; TNF-α: tumor necrosis factor alpha; IL-6: interleukin 6; NLR: neutrophil-lymphocyte ratio; MLR: monocyte-lymphocyte ratio; PLR: platelet-lymphocyte ratio; AISI: aggregate index of systemic inflammation; SII: systemic immune-inflammation index; MCVL: ratio between the mean corpuscular volume/lymphocytes; IIC: cumulative inflammatory index; PNI: prognostic nutritional index; SD: standard deviation; * *p* < 0.05: statistically significant; **: reached the significance limit.

**Table 7 ijms-26-03661-t007:** Diagnostic performance of the investigated parameters.

Parameter	AUC	Std. Error	Cut-Off Values	95% CI	Sensitivity %	Specificity %	Youden Index	*p*-Value
IL-6	0.866	0.045	40.30	0.778–0.954	100.00	60.00	0.600	<0.0001
PTX3	0.720	0.065	1888	0.593–0.846	67.60	73.30	0.409	0.003
MCVL	0.677	0.064	39.60	0.560–0.795	63.30	66.70	0.300	0.047
TNF-α	0.671	0.068	222	0.538–0.803	67.60	56.70	0.243	0.019
hs-CRP	0.635	0.071	1025	0.497–0.774	61.80	56.70	0.185	0.036
PLR	0.616	0.060	101	0.491–0.740	73.30	60.00	0.333	0.006
SII	0.588	0.064	397	0.463–0.713	56.70	60.00	0.167	0.173
NLR	0.586	0.068	1.98	0.453–0.718	65.00	60.00	0.250	0.187
MLR	0.579	0.063	0.212	0.456–0.702	53.30	56.70	0.100	0.223
IIC	0.572	0.066	2.35	0.442–0.702	61.70	60.00	0.217	0.027
AISI	0.571	0.065	233	0.443–0.698	56.70	53.30	0.100	0.277

PTX3: pentraxin 3; hs-CRP: high-sensitivity C-reactive protein; TNF-α: tumor necrosis factor alpha; IL-6: interleukin 6; NLR: neutrophil-lymphocyte ratio; MLR: monocyte-lymphocyte ratio; PLR: platelet-lymphocyte ratio; AISI: aggregate index of systemic inflammation; SII: systemic immune-inflammation index; MCVL: ratio between the mean corpuscular volume/lymphocytes; IIC: cumulative inflammatory index.

## Data Availability

The data used to support the findings of this study are available from the corresponding author upon reasonable request.

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
