# Peer review of "Association Between Pentraxins and Obesity in Prediabetes and Newly Diagnosed Type 2 Diabetes Mellitus Patients"

_ijms, 2025, doi:10.3390/ijms26083661_

Round 1
Reviewer 1 Report
Comments and Suggestions for Authors
The MS identifies PTX3 and MCVL as promising biomarkers for early inflammation detection in T2DM, adding new insights into obesity-related systemic inflammation. However, methodological limitations and statistical concerns require significant refinement.
Major issue:
There are no simply contorl subjects in this study.
A key limitation of this study is the absence of a healthy control group, which restricts the ability to determine whether the observed elevations in PTX3, IL-6, and other inflammatory markers are specific to diabetes progression or merely reflect general metabolic dysfunction. Without a normoglycemic reference, it remains unclear whether these biomarkers serve as early indicators of diabetes risk or are already elevated in metabolically healthy individuals with obesity.
Additionally, while ROC analysis differentiates PreDM from T2DM, its clinical utility is limited without assessing whether these biomarkers can predict the transition from normoglycemia to PreDM. The correlation between PTX3 and BMI, for example, may indicate general adipose tissue inflammation rather than a diabetes-specific process. Similarly, the observed decline in MCVL with increasing HbA1c suggests a metabolic association, but its specificity remains uncertain.
Future studies should include a three-group comparison (normoglycemia, PreDM, and T2DM) to validate the diagnostic and predictive value of these biomarkers. Longitudinal analyses are also needed to determine whether these markers precede diabetes onset or arise as secondary effects of hyperglycemia.
Minor issues:
1. The title is too long. A more concise version is needed.
2. ROC analysis results should include AUC values in the abstract. Clearly state statistical values (p-values, AUC, sensitivity/specificity).
3. Clearly state the hypothesis at the end of the introduction.
4. No justification for sample size selection (n = 90).
5. Confounding factors (diet, medication) are not controlled.
6. ROC analysis lacks confidence intervals (CI) for AUC values.
The English could be improved to more clearly express the research.
Author Response
Author's Reply to the Review Report (Reviewer 1)
Dear Reviewer,
Thank you very much for taking the time to analyze our manuscript, and for your kind appreciation and valuable suggestions.
All the typing recommended changes were performed in the body of our manuscript, with the Track Changes function activated.
Comments and Suggestions for Authors
The MS identifies PTX3 and MCVL as promising biomarkers for early inflammation detection in T2DM, adding new insights into obesity-related systemic inflammation. However, methodological limitations and statistical concerns require significant refinement.
Major issue:
Comments 1: There are no simply contorl subjects in this study. A key limitation of this study is the absence of a healthy control group, which restricts the ability to determine whether the observed elevations in PTX3, IL-6, and other inflammatory markers are specific to diabetes progression or merely reflect general metabolic dysfunction. Without a normoglycemic reference, it remains unclear whether these biomarkers serve as early indicators of diabetes risk or are already elevated in metabolically healthy individuals with obesity.
Response 1: When we designed the study, we aimed to investigate newly diagnosed patients with T2DM and compare the results with a group of prediabetic patients, as a control group. Also, from personal experience, when we determined the values ​​of PTX3, TNF and IL-6 in the healthy control group, they were low, therefore, when we compared them with values ​​from pathological groups, we found these reports to be inconclusive.
Comments 2: Additionally, while ROC analysis differentiates PreDM from T2DM, its clinical utility is limited without assessing whether these biomarkers can predict the transition from normoglycemia to PreDM. The correlation between PTX3 and BMI, for example, may indicate general adipose tissue inflammation rather than a diabetes-specific process. Similarly, the observed decline in MCVL with increasing HbA1c suggests a metabolic association, but its specificity remains uncertain.
Response 2: We respect your opinions. Given that these associations have not been previously reported, we considered them relevant and wanted to publish them.
Comments 3: Future studies should include a three-group comparison (normoglycemia, PreDM, and T2DM) to validate the diagnostic and predictive value of these biomarkers. Longitudinal analyses are also needed to determine whether these markers precede diabetes onset or arise as secondary effects of hyperglycemia.
Response 3: Thank you for the recommendation. We will continue the study and include a considerable healthy control group.
Minor issues:
Comments 4: 1. The title is too long. A more concise version is needed.
Response 4: Revised according to the recommendations made.
Comments 5: 2. ROC analysis results should include AUC values in the abstract. Clearly state statistical values (p-values, AUC, sensitivity/specificity).
Response 5: Revised according to the recommendations made.
Comments 6: 3. Clearly state the hypothesis at the end of the introduction.
Response 6: We specified our study objectives/hypotheses at the end of the introduction.
This retrospective study focused on evaluating immune and inflammatory status by determining serum levels of PTX3, hs-CRP, TNF-α, and IL-6, as well as hematological markers (NLR, MLR, PLR, AISI, SII, MCVL, and IIC) within a cohort of patients diagnosed with preDM and newly diagnosed T2DM. This cohort was selected using data from two university clinical hospitals representing Dolj County in Romania. Additionally, we aimed to assess the association of these parameters with clinical and biochemical measures, along with various obesity-related indices, to identify potential correlations among them. The third major aim of the study was to evaluate the diagnostic accuracy of PTX3, hs-CRP, TNF-α, IL-6, NLR, MLR, PLR, AISI, SII, MCVL, and IIC as distinct early predictive factors and to ascertain the inflammatory status in patients both with and with-out T2DM, using receiver operating characteristic (ROC) curve analysis.
Comments 7: 4. No justification for sample size selection (n = 90).
Response 7: This was the group of patients, over a six-month period, who sought a specialist consultation and who remained after we applied established inclusion and exclusion criteria.
Comments 8: 5. Confounding factors (diet, medication) are not controlled.
Response 8: We mentioned in 4.3. Medical background, evaluation of biometric parameters, and demographic information:
Regarding the patients' medication, some patients were on treatment with antihypertensives (perindopril, perindopril/indapamide combination), statins (atorvastatin, rosuvastatin).
Regarding the dietary factor, most patients did not have a diet established by a specialist. Also, following the consultation, it was noted that the subjects did not perform physical activity more than 4 days per week and were considered sedentary.
Comments 9: 6. ROC analysis lacks confidence intervals (CI) for AUC values.
Response 9: Revised according to the recommendations made.
Reviewer 2 Report
Comments and Suggestions for Authors
1. Some tables (such as Table 1 and Table 2) have disorderly numerical alignment and inconsistent unit labeling (such as HbA1c in% and other indicators in pg/mL or mmol/L). Unified format is required to enhance readability.
2. Some abbreviations (such as preDM and MCVL) are not fully defined when they first appear, and complete terminology needs to be added (for example, "prediabetes" should be abbreviated as "preDM" instead of "preDM").
3. The discussion section should focus more on the core findings (such as the association between PTX3 and obesity indicators, and the diagnostic value of MCVL) to avoid lengthy and repetitive results. Suggest summarizing the innovation and limitations in points.
4. The study is a retrospective design with selection bias (such as excluding patients with microvascular complications), which may limit the generalizability of the results. Suggest emphasizing this limitation in the discussion section and suggesting future prospective research.
5. Some indicators (such as MCVL and IIC) were not clearly defined when they first appeared in the abstract, and the literature supporting their use as novel inflammatory indicators was not fully cited (only citing the author's team's previous research [34]). More external validation studies are needed to enhance its scientific validity.
6. Although the AUC of IL-6 is high (86.6%), its specificity is only 60%, which may lack the specificity of diabetes due to its extensive participation in inflammatory reaction. Further discussion is needed on its practical application value in clinical practice, such as whether it needs to be combined with other indicators.
Author Response
Author's Reply to the Review Report (Reviewer 2)
Dear Reviewer,
Thank you very much for taking the time to analyze our manuscript, and for your kind appreciation and valuable suggestions.
All the typing recommended changes were performed in the body of our manuscript, with the Track Changes function activated.
Comments and Suggestions for Authors
Comments 1:
- Some tables (such as Table 1 and Table 2) have disorderly numerical alignment and inconsistent unit labeling (such as HbA1c in% and other indicators in pg/mL or mmol/L). Unified format is required to enhance readability.
Response 1: Thank you for the recommendation. The revisions have been made based on the recommendations provided.
Comments 2:
- Some abbreviations (such as preDM and MCVL) are not fully defined when they first appear, and complete terminology needs to be added (for example, "prediabetes" should be abbreviated as "preDM" instead of "preDM").
Response 2: We respect your opinions, but we don’t understand your recomandation : for example, "prediabetes" should be abbreviated as "preDM" instead of "preDM". Probably you wanted to mention PreDM. We have revised the manuscript and made the corrections.
Comments 3:
- The discussion section should focus more on the core findings (such as the association between PTX3 and obesity indicators, and the diagnostic value of MCVL) to avoid lengthy and repetitive results. Suggest summarizing the innovation and limitations in points.
Response 3: The revisions have been made based on the recommendations provided.
Comments 4:
- The study is a retrospective design with selection bias (such as excluding patients with microvascular complications), which may limit the generalizability of the results. Suggest emphasizing this limitation in the discussion section and suggesting future prospective research.
Response 4: Thank you for the recommendation. The revisions have been made based on the recommendations provided
Comments 5:
- Some indicators (such as MCVL and IIC) were not clearly defined when they first appeared in the abstract, and the literature supporting their use as novel inflammatory indicators was not fully cited (only citing the author's team's previous research [34]). More external validation studies are needed to enhance its scientific validity.
Response 5: The revisions have been made based on the recommendations provided. In addition to reference 34, we also mentioned references 35 and 36, to which we appended the most recent published papers on MCVL and IIC, respectively (citation 37). There are currently no other manuscripts pertaining to MCVL and IIC.
Comments 6:
- Although the AUC of IL-6 is high (86.6%), its specificity is only 60%, which may lack the specificity of diabetes due to its extensive participation in inflammatory reaction. Further discussion is needed on its practical application value in clinical practice, such as whether it needs to be combined with other indicators.
Response 6: Thank you for the recommendation. Taking into account the recommendations in comments 3, which were also made by another reviewer, we did not discuss IL-6 or TNF-α further.
Reviewer 3 Report
Comments and Suggestions for Authors
The study mainly explores the association between inflammatory biomarkers (PTX3, hs-CRP, TNF-α, IL-6), hematological indicators (NLR, MLR, PLR, etc.) and obesity, and evaluates the value of these indicators in disease diagnosis. Overall, it is well organized but needs some necessary revisions.
- The study included only 90 patients (60 with type 2 diabetes and 30 with prediabetes), which is a relatively small sample size for correlation study. This may limit the generalizability of the findings.
- Long-term follow-up data may be preferred to determine the dynamic relationship between changes in inflammatory biomarkers and the progression of diabetes.
- The figure and table formats should be consistent with the requirements of this journal. Significance analysis should be conducted for figure 1.
- The discussion section should focus on the contributions of current study.
- The reference format should be consistent. For instance, ref. 81&85 has a different format.
- The paper title and abstract should be improved. High-Sensitivity C-Reactive Protein? C-Reactive Protein?
The English could be improved to more clearly express the research.
Author Response
Author's Reply to the Review Report (Reviewer 3)
Dear Reviewer,
Thank you very much for taking the time to analyze our manuscript, and for your kind appreciation and valuable suggestions.
All the typing recommended changes were performed in the body of our manuscript, with the Track Changes function activated.
Comments and Suggestions for Authors
The study mainly explores the association between inflammatory biomarkers (PTX3, hs-CRP, TNF-α, IL-6), hematological indicators (NLR, MLR, PLR, etc.) and obesity, and evaluates the value of these indicators in disease diagnosis. Overall, it is well organized but needs some necessary revisions.
Comments 1:
The study included only 90 patients (60 with type 2 diabetes and 30 with prediabetes), which is a relatively small sample size for correlation study. This may limit the generalizability of the findings.
Response 1: This was the group of patients, over a six-month period, who sought a specialist consultation and who remained after we applied established inclusion and exclusion criteria.
Also, we recognize that our study has inherent limitations at the end of the Discussion Section.
Comments 2:
Long-term follow-up data may be preferred to determine the dynamic relationship between changes in inflammatory biomarkers and the progression of diabetes.
Response 2: Thank you for the recommendation. We want to continue the study to determine the dynamic relationship between changes in inflammatory biomarkers and the progression of diabetes.
Comments 3:
The figure and table formats should be consistent with the requirements of this journal. Significance analysis should be conducted for figure 1.
Response 3: Revised according to the recommendations made.
Comments 4:
The discussion section should focus on the contributions of current study.
Response 4: We have revised the manuscript and made the corrections.
Comments 5:
The reference format should be consistent. For instance, ref. 81&85 has a different format.
Response 5: We have checked the references and made the corrections when necessary.
Comments 6:
The paper title and abstract should be improved. High-Sensitivity C-Reactive Protein? C-Reactive Protein?
Response 6: Revised according to the recommendations made.
Comments 7:
Comments on the Quality of English Language
The English could be improved to more clearly express the research.
Response 7: Revised according to the recommendations made.
Reviewer 4 Report
Comments and Suggestions for Authors
The present research provides valuable insights into the relationship between inflammatory biomarkers and prediabetes or diabetes and compares the power of prediction of inflammatory status. Underlying these interesting results, it is evident that there is a lot of work. However, some minor points need to be addressed for better clarity.
First of all, the abstract is very long, while it should be a maximum of 200 words. Reducing it to 200 words will help readers understand the paper quickly if it is of interest to them.
Second, the inclusion criteria are a bit confusing. The article compares inflammatory markers in prediabetes and type 2 diabetes, but the inclusion section says that patients were included in the study if they had type 2 diabetes.
At the end of the introduction, the study has three main objectives. Please state why this study would bring knowledge to the field.
The figures are interesting, but insufficiently explained in the text.
The conclusion could be improved to be more concise and compelling to summarize the findings and their applicability.
Author Response
Author's Reply to the Review Report (Reviewer 4)
Dear Reviewer,
Thank you very much for taking the time to analyze our manuscript, and for your kind appreciation and valuable suggestions.
All the typing recommended changes were performed in the body of our manuscript, with the Track Changes function activated.
Comments and Suggestions for Authors
The present research provides valuable insights into the relationship between inflammatory biomarkers and prediabetes or diabetes and compares the power of prediction of inflammatory status. Underlying these interesting results, it is evident that there is a lot of work. However, some minor points need to be addressed for better clarity.
Comments 1:
First of all, the abstract is very long, while it should be a maximum of 200 words. Reducing it to 200 words will help readers understand the paper quickly if it is of interest to them.
Response 1: Thank you for the recommendation. The revisions have been made based on the recommendations provided.
Comments 2:
Second, the inclusion criteria are a bit confusing. The article compares inflammatory markers in prediabetes and type 2 diabetes, but the inclusion section says that patients were included in the study if they had type 2 diabetes.
Response 2: The revisions have been made based on the recommendations provided.
Comments 3:
At the end of the introduction, the study has three main objectives. Please state why this study would bring knowledge to the field.
Response 3: Revised according to the recommendations made.
Comments 4:
The figures are interesting, but insufficiently explained in the text.
Response 4: Revised according to the recommendations made.
Comments 5:
The conclusion could be improved to be more concise and compelling to summarize the findings and their applicability.
Response 5: The revisions have been made based on the recommendations provided.
Round 2
Reviewer 1 Report
Comments and Suggestions for Authors
Thank you for Q1 of your clarification. To enhance the interpretation of your results, please consider providing the relevant data on PTX3, TNF, and IL-6 levels in healthy individuals as supplementary material.
Author Response
Dear Reviewer,
We would like to express our sincere gratitude for the time and effort you devoted to reviewing our manuscript. Your thoughtful evaluation, encouraging remarks, and constructive suggestions are deeply appreciated. Your feedback has been invaluable in helping us improve the quality and clarity of our work, and we are truly grateful for your insightful contributions to the development of this manuscript.
Comments 1: To enhance the interpretation of your results, please consider providing the relevant data on PTX3, TNF, and IL-6 levels in healthy individuals as supplementary material.
Response 1: Based on our prior experience, the measured levels of PTX3, TNF, and IL-6 in the healthy control group consistently remained low. Consequently, when these values were compared with those obtained from pathological groups, the differences did not yield statistically or clinically conclusive results. This limited the interpretative value of such comparisons and influenced our decision to focus our analysis accordingly.
Reviewer 2 Report
Comments and Suggestions for Authors
The author has addressed my concerns
Author Response
Dear Reviewer,
We would like to express our sincere gratitude for the time and effort you devoted to reviewing our manuscript. Your thoughtful evaluation, encouraging remarks, and constructive suggestions are deeply appreciated. Your feedback has been invaluable in helping us improve the quality and clarity of our work, and we are truly grateful for your insightful contributions to the development of this manuscript.
Reviewer 3 Report
Comments and Suggestions for Authors
I have no further comments.
Comments on the Quality of English LanguageThe English could be improved to more clearly express the research.
Author Response

(The authors gave the same response as above.)
